# Protective Effect of the Novel Melatonin Analogue Containing Donepezil Fragment on Memory Impairment via MT/ERK/CREB Signaling in the Hippocampus in a Rat Model of Pinealectomy and Subsequent Aβ_1-42_ Infusion

**DOI:** 10.3390/ijms25031867

**Published:** 2024-02-03

**Authors:** Jana Tchekalarova, Petya Ivanova, Desislava Krushovlieva, Lidia Kortenska, Violina T. Angelova

**Affiliations:** 1Institute of Neurobiology, Bulgarian Academy of Sciences, 1113 Sofia, Bulgaria; ivanova.petya91@gmail.com (P.I.); daisyveko@gmail.com (D.K.); lidiakortenska@gmail.com (L.K.); 2Department of Chemistry, Faculty of Pharmacy, Medical University of Sofia, 1000 Sofia, Bulgaria; violina_stoyanova@abv.bg

**Keywords:** Alzheimer’s disease, melatonin, melatonin analogue, *icv*Aβ_1-42_, pTAU, MT receptors, ERK, CREB

## Abstract

A reduction in melatonin function contributes to the acceleration of Alzheimer’s disease (AD), and understanding the molecular processes of melatonin-related signaling is critical for intervention in AD progression. Recently, we synthesized a series of melatonin analogues with donepezil fragments and tested them in silico and in vitro. In this study, one of the most potent compounds, **3c**, was evaluated in a rat model of pinealectomy (pin) followed by icvAβ_1-42_ infusion. Melatonin was used as the reference drug. Treatment with melatonin and **3c** (10 mg/kg, i.p. for 14 days) had a beneficial effect on memory decline and the concomitant increase in hippocampal Aβ_1-42_ and pTAU in the pin+icvAβ_1-42_ rats. Melatonin supplementation facilitated non-amyloidogenic signaling via non-receptor (histone deacetylase sirtuin 1, SIRT1) and receptor-related signaling (MT/ERK/CREB). The hybrid **3c** analogue up-regulated the MT_1A_ and MT_2B_ receptors, pERK and pCREB. Our results strongly support the hypothesis that melatonin-related analogues may become a promising drug candidate for Alzheimer’s disease therapy.

## 1. Introduction

Alzheimer’s disease (AD) is a progressive brain disorder characterized by severe dementia that affects approximately 10% of the world’s elderly population [1]. The accumulation of extracellular amyloid plaques, composed mainly of amyloid (A)β, and the intracellular formation of hyperphosphorylated TAU protein (pTAU) are considered the unique hallmarks of AD pathogenesis. The imbalance between Aβ production and Aβ clearance with the formation of neurotoxic Aβ_1-40_ and Aβ_1-42_ fragments, leading to an increase in oxidative stress in the brain, is one of the accepted hypotheses for the etiology of AD. The accompanying behavioral disturbances include circadian rhythm disruption, sleep disturbance, impaired cognitive function and memory loss. Today, the priority in AD research is to discover the signaling pathways that are closely associated with predisposition to the development of AD pathogenesis. This issue is a crucial step in finding new targets to manipulate and thus control the progression of this neurological disease. Although there is considerable funding for research into practical therapeutic approaches, there is still no approved drug with the three “golden” characteristics needed to reach the market, i.e., (i) to be able to suppress neurodegeneration and disease progression; (ii) to have no side effects in chronic treatment and (iii) to be inexpensive.

Experimental and clinical evidence suggests that the circadian dysregulation of the sleep–wake cycle is part of the early symptoms of the disease predisposing to memory deficits and AD progression [2]. The treatment of sleep–wake cycle disruption at the onset or before the onset of AD symptoms has been suggested to reduce or even prevent pathogenesis development [3]. For example, suvorexant, an orexin receptor antagonist, was recently approved by the Food and Drug Administration in the US for the treatment of sleep–wake circadian disruption in the early stages of AD [4]. The beneficial role of melatonin in AD pathogenesis has also been extensively studied [5] due to its primary function in regulating the circadian rhythms in the body, including synchronizing the sleep–wake cycle and promoting sleep. Melatonin is released from the pineal gland during the dark phase when the inhibitory control from the suprachiasmatic nucleus is weakened by exposure to light.

The main points that point to a close link between the melatoninergic system and AD pathogenesis are as follows: (1) disturbed circadian rhythms (melatonin plays a key role in the resynchronization of circadian rhythms); (2) disturbed pro-oxidant/antioxidant balance (melatonin is a potent antioxidant and free radical scavenger); and (3) neuroinflammation (the hormone has anti-inflammatory activity). Regarding the role of the melatonin system in the three main hallmarks of AD, i.e., Aβ plaques formation, p-TAU, and cholinergic system dysfunction, a growing body of evidence points to a putative neuroprotective effect of the hormone [6,7]. There is ongoing research to explore its therapeutic potential and the development of melatonin-related compounds for treating AD pathogenesis. However, the underlying mechanism associated with the beneficial effect of the melatonin system is still an area of ongoing research.

It is hypothesized that melatonin deficiency and pineal gland dysfunction, in particular, may be critical triggers that predispose to AD pathogenesis [8]. Specifically, patients with AD had diminished function of this gland due to a reduction in the size and calcification [9,10], which decreased melatonin levels in the cerebrospinal fluid (CSF), serum, and urine [8,11,12,13].

The pineal ablation gland in rodents has been accepted as a relevant model to study the detrimental effects of melatonin deficiency, which resemble the signaling dysfunction associated with neurodegenerative diseases, including AD [14]. Recently, we reported that simultaneous removal of the pineal gland and intracerebroventricular (icv) infusion of Aβ_1-42_ exacerbated behavioral responses and concomitant oxidative stress in the hippocampus and the frontal cortex [15]. Our research team synthesized and tested a series of new melatonin analogues and sulfonyl hydrazone compounds with a melatonin scaffold [16]. Docking analysis revealed a plausible mechanism of action for one of the most potent compounds, the **3c**, which is associated with melatonin/MT/receptors and the inhibition of acetylcholinesterase (AchE) and butyrylcholinesterase (BchE). In the present study, the most effective and potent compound, **3c**, with low in vitro neurotoxicity, was selected and tested in a rat model of melatonin deficiency induced by pinealectomy (pin) followed by *icv*Aβ_1-42_ infusion. Melatonin was used as the referent drug. The underlying protective mechanism of the new compound was determined, including the possible signaling pathway closely related to the melatonin system and involved in the suppression of Aβ_1-42_ accumulation in the hippocampus. We hypothesized that this novel melatonin analogue, **3c**, behaves as a potent MT receptor agonist promoting the non-amyloidogenic pathway by facilitating the MT receptor-related ERK/CREB signaling process. The timeline of the experimental steps is shown in Figure 1.

## 2. Results

### 2.1. The Circadian Pattern of Motor Activity and Effects of Melatonin and ***3c*** Compound in a Rat Model of Pinealectomy and icvAβ_1-42_ Infusion

The novel melatonin analogue, 3c, containing a donepezil fragment, was synthesized by a condensation reaction of a 1-benzylpiperidine-4-carbaldehyde and 1H-Indole-5-carbohydrazide, at a molar ratio of 1:1, in abs. ethanol for 1–2 h [16] (Figure 2).

The 24 h registration of the horizontal and vertical activity of the control group (sham-veh-veh group), model group (pin+Aβ_1-42_-veh), positive control (pin+Aβ_1-42_-mel) and treatment group (pin+Aβ_1-42_-**3c**) was performed in the actimeter. All groups showed a circadian pattern of motor activity, which was also confirmed by the cosinor analysis (Figure 3A,B; Table 1).

The horizontal and vertical activity, total distance and mean velocity, measured in the light and the dark phases, respectively, were also evaluated (Figure 4A–D). The daily pattern of activity was verified for all groups (*p* < 0.05, *p* < 0.001, dark vs. light phase). The model group was characterized by an increased horizontal activity compared to the control group (*p* < 0.05), as well as increased vertical activity compared to the control group (*p* < 0.001) (Figure 4A,B). Both drugs tended to compensate for the pin+Aβ_1-42_-induced changes in horizontal activity (Figure 4A). At the same time, only the melatonin treatment attenuated the enhanced effect of pin+Aβ_1-42_ on vertical activity (*p* < 0.001 vs. pin+Aβ_1-42_-veh group) (Figure 4B). There was no difference in the total distance and velocity among groups (*p* > 0.05) (Figure 4C,D).

### 2.2. Effects of Melatonin and ***3c*** Compound on Memory Impairment Induced by Pinealectomy and icvAβ_1-42_ Infusion

To evaluate the efficacy of the new hybrid compound, **3c**, and melatonin, used as a reference to compensate for the pinealectomy-induced melatonin deficiency and the Aβ_1-42_-related memory impairment, we performed a battery of cognitive tests, including the Object recognition test (ORT), the Object location test (OLT) and the Y-maze test.

#### 2.2.1. Object Recognition Test

The ORT was used to assess the cognitive abilities related to the natural curiosity of rodents to explore novel objects. The effects of melatonin and **3c** in rats with pinealectomy and icvAβ_1-42_ infusion on short-term novelty recognition memory were evaluated. The vehicle-treated pin+Aβ_1-42_ group significantly reduced the Discrimination index (DI) for both time (*p* < 0.05 vs. sham-veh-veh group) and counts (*p* < 0.05 vs. sham-veh-veh group) (Figure 5A,B). While the reference drug significantly alleviated DI (count) vs. pin+Aβ_1-42_-veh group (*p* < 0.05), the sub-chronic treatment with melatonin or **3c** showed a tendency to reverse the impaired memory recognition ((*p* > 0.05 vs. sham-veh-veh group) and (Figure 5B)).

#### 2.2.2. Object Location Test

The OLT was used to assess the effect of melatonin and **3c** on pinealectomy-induced and icvAβ_1-42_ infusion-induced deficits in short-term spatial memory. For this purpose, the spontaneous response of rats to preferentially explore the novel object placed in an open field, instead of a familiar object placed in the same position, as in the sample phase, (see Section 4) was measured. As in the ORT, the pin+Aβ_1-42_-veh group had a lower DI than the control group (*p* < 0.001), indicating impaired short-term spatial memory (Figure 6). Although treatment with the hybrid compound **3c** showed a significantly lower DI than the sham-veh-veh group (*p* < 0.05), both melatonin and **3c** reversed the pin+Aβ_1-42_-induced memory impairment (*p* < 0.001 and *p* < 0.05 vs. pin-Aβ_1-42_-veh group, respectively).

#### 2.2.3. Y-Maze Test

Spontaneous alternation behavior (SAB) was assessed in the first trial of the Y-maze test used to evaluate spatial memory. Lower SAB was found in the pin-Aβ_1-42_-veh group compared to the sham-veh-veh group (*p* < 0.001) (Figure 7A). Both the reference group melatonin and the hybrid compound **3c** alleviated the impaired SAB (*p* < 0.05 and *p* < 0.001 vs. pin-Aβ_1-42_-veh group, respectively).

Short-term spatial memory was also assessed in the Y maze second trial. As in the OLT, the melatonin deficit model and subsequent *icv* Aβ_1-42_ infusion reduced DI (time) and DI (count) (*p* < 0.001 vs. control) (Figure 7B,C), respectively. Treatment with both the positive control melatonin and compound **3c** reduced the pin+*icv* Aβ_1-42_-induced impairment of spatial memory (DI (time): *p* < 0.001, pin-Aβ_1-42_-mel group vs. pin-Aβ_1-42_-veh group; *p* < 0.05, pin-Aβ_1-42_-**3c** group vs. pin-Aβ_1-42_-veh group (Figure 7B); and DI (count): *p* < 0.01, pin-Aβ_1-42_-veh group vs. pin-Aβ_1-42_-mel and pin-Aβ_1-42_-**3c** group, respectively, (Figure 7C). However, the treatment with compound **3c** was unable to correct the pin-Aβ_1-42_-induced reduction in DI (time) (*p* < 0.001 vs. sham-veh-veh group) (Figure 7B).

### 2.3. The Expression of Markers of ADs and Effects of Melatonin and Compound ***3c*** in a Rat Model of Pinealectomy and icvAβ_1-42_ Infusion

The effects of sub-chronic treatment with the positive control melatonin and the hybrid compound **3c** on pathological markers closely associated with ADs, including the expression of Aβ_1-42_, pTAU protein and the hippocampal AchE level, were evaluated following the induction of melatonin deficiency and concomitant icvAβ_1-42_ infusion one week after pinealectomy. The hippocampus procedure isolation was performed 24 h after the last memory test (see Figure 1).

The expression of Aβ_1-42_ and pTAU in the hippocampus was significantly increased in the model group treated with a vehicle compared to the control group (*p* < 0.05 and *p* < 0.05, respectively) (Figure 8A,B). At the same time, there was no change in the level of AchE among the four groups (Figure 8C). Treatment with melatonin and compound **3c** failed to reverse the model-induced elevation of AD markers (*p* > 0.05 vs. control group).

### 2.4. Neuroprotective Effect of Melatonin and the Hybrid Compound ***3c*** on Aβ_1-42_ Neurotoxicity via MT Receptors/ERK/CREB Signaling

Previous studies have shown that melatonin exerts neuroprotection against Aβ neurotoxicity by redirecting the APP metabolism to the non-amyloidogenic pathways. In the present study, we investigated the possible involvement of this molecular mechanism in the beneficial effects of melatonin and melatonin-related compound **3c** on memory decline in the pin+Aβ_1-42_ rat model. *W*e measured the levels of several signaling molecules associated with the non-amyloidogenic pathway, including the expression of the two receptors MT_1A_ and MT_2B_, respectively, sirtuin 1 (SIRT1), pERK1/2 and pCREB.

The effects of sub-chronic treatment with the positive control melatonin and the hybrid compound **3c** on the signaling molecules associated with the non-amyloidogenic pathway activated via MT_1A_/MT_2B_ receptor subtypes in the hippocampus were assessed. The pinealectomy model, followed by icvAβ_1-42_ infusion, down-regulated the two MT receptor subtypes in the hippocampus compared to the sham+veh-veh group (*p* < 0.05 and *p* < 0.01, respectively) (Figure 9A,B). Treatment with the positive control melatonin and compound **3c** reversed the pin+icvAβ_1-42_-related decreased expression in the MT_1A_ receptor subtype (*p* < 0.05 vs. control group), whereas only the novel melatonin-related hybrid up-regulated the pin-Aβ decrease in the MT_2B_ receptor subtype (*p* < 0.01 vs. pin-Aβ_1-42_-veh group) in the hippocampus. Interestingly, melatonin failed to correct the pin-Aβ_1-42_-related down-regulation of the MT_2B_ receptor subtype in the hippocampus (*p* < 0.05, vs. pin-Aβ_1-42_-veh group) (Figure 9B).

Furthermore, the compound **3c** reversed the pin-Aβ_1-42_-induced decreased expression of both the pERK1/2 (*p* = 0.01 vs. pin-Aβ_1-42_-veh group) and pCREB in the hippocampus (*p* < 0.05 vs. pin-Aβ_1-42_-veh group) (Figure 10A,B). At the same time, the positive effect of melatonin on these signaling molecules was even higher (*p* < 0.001 vs. pin-Aβ_1-42_-veh group, Figure 10A and *p* < 0.05 vs. sham-veh-veh and pin-Aβ_1-42_-veh group, Figure 10B, respectively). Similarly, melatonin treatment significantly increased the SIRT1 in the hippocampus (*p* < 0.01 vs. sham-veh-veh group, *p* < 0.001 vs. pin-Aβ_1-42_-veh group), while the hybrid compound **3c** was ineffective (*p* > 0.05 vs. pin-Aβ_1-42_-veh group; *p* = 0.01 vs. pin-Aβ_1-42_-mel group) (Figure 10C).

## 3. Discussion

Recently, our team designed, synthesized and characterized a series of melatonin-based hybrid compounds with hydrazine scaffolds to determine whether they have the potential for further evaluation in AD therapy [16]. Based on in silico and in vitro data, one of the two lead compounds, **3c**, was selected for further in vivo study in a rat model of pinealectomy followed by *icv*Aβ_1-42_ infusion. Melatonin was used as a positive control. The potential protective effect of the hybrid compound **3c** against the Aβ protein-related neurotoxicity and a plausible underlying mechanism were explored. The hypothesis that melatonin and the compound **3c** could stimulate non-amyloidogenic signaling in the hippocampus via activity on MT_1A_/MT_2B_ receptors in the pin+*icv*Aβ_1-42_ model was based on the recently reported potency of the novel hybrid compound to bind to the two MT receptor subtypes by molecular docking analysis [16]. Our results showed that, while the beneficial effect of the positive control melatonin on AD-related pathogenesis involves the stimulation of the non-amyloidogenic pathway via both non-receptor (SIRT1) and MT_1A_-receptor-related ERK1/2/CREB signaling, the novel analogue **3c** only attenuated memory decline via the MT_1A_/MT_2B_ receptor-related ERK1/2/CREB signaling in a rat model of pin+icvAβ_1-42_. The schematic presentation of the proposed mechanism underlying the effects of melatonin and the compound **3c** is shown in Figure 1.

We previously reported that the simultaneous induction of melatonin deficiency by pinealectomy and AD-related pathogenesis by *icv* infusion of Aβ_1-42_ in rats induces typical AD behavioral symptom changes such as increased anxiety and cognitive impairment [15]. Demir et al. (2017) and Zhu et al. (2004) reported that melatonin deficiency exacerbated AD-like changes and memory impairment [17,18].

The idea that CSF melatonin levels decrease in the preclinical stages of AD when patients do not show cognitive impairment [8] suggests that melatonin deficiency may be a critical factor predisposing to the development AD. In the present study, we modified the model previously reported by our team [15] in order to simulate the preclinical stage of AD with a drop in melatonin blood levels induced by pinealectomy to assess the efficacy of advanced prophylactic treatment with the hormone and the novel melatonin hybrid compound **3c** on AD pathogenesis induced by a subsequent *icv*Aβ_1-42_-infusion. Long-term supplementation with a high dose of melatonin (50 mg/kg for 40 days) reversed behavioral impairments and the associated increase in oxidative stress in the frontal cortex and the hippocampus [15]. Melatonin may play a role in the modulating of spatial learning and memory, and its potential effects on these cognitive functions in AD models have been investigated in several studies [19,20,21,22]. The beneficial effects of the melatonin system on spatial memory may be related to its ability to regulate disrupted circadian rhythms, neuroprotection, antioxidant and anti-inflammatory properties. The hippocampus is known to be a brain region critical for spatial learning and memory. Melatonin receptors are present in the hippocampus [23] and, so, the hormone could affect synaptic plasticity and the processes closely linked to memory formation in this region.

In addition, the modulatory role of melatonin on acetylcholine [7], the neurotransmitter that plays a crucial role in cognition [24], is also an essential factor suggesting the beneficial effects of melatonin on spatial memory. Our results showed that, while the novel hybrid compound **3c** is ineffective against impaired non-spatial cognitive performance, such as the ORT in a rat model of pin+*icv*Aβ_1-42_, this melatonin analogue has a comparable potency to the positive control melatonin in correcting the impaired short-term and spatial hippocampus-dependent memory. The beneficial role of melatonin supplementation on spatial memory decline has been demonstrated in several models of AD, including the hereditary form of AD, such as mouse transgenic and knockout mice [19,25,26], and the sporadic form of AD, such as senescence-accelerated OXYS rats [21], streptozotosin- [27,28] and *icv*Aβ_1-42_-induced models [29].

There is sufficient evidence in the literature to support the idea that the soluble Aβ oligomers and the total Aβ levels, in particular, are responsible for progressive memory decline in AD [30,31], suggesting that these factors may be more important than the formation of plaques in understanding the pathology of AD. Both experimental reports using transgenic APP mice [32,33] and human brain tissue [30,31] have shown that cognitive changes occur before the formation of plaques, supporting the notion that soluble forms of Aβ may cause cognitive impairment. In addition, soluble Aβ oligomers, containing Aβ_1–40_ or Aβ_1-42_, detected in AD patients are thought to contribute more to the neurotoxicity associated with AD than Aβ deposition. Thereby, interventions aimed at reducing or preventing the neurotoxicity associated with the soluble Aβ oligomers may be crucial for the sequential treatment of AD pathogenesis. To verify the model of melatonin deficiency induced by pinealectomy followed by *icv*Aβ_1-42_ infusion, we first examined the expression of Aβ_1-42_, pTAU and AchE levels in the hippocampus. The observed increase in Aβ_1-42_ is consistent with our previous report, in which the pineal gland was removed at the same time as the toxic oligomer infusion [15].

The present study also confirmed that pTAU was significantly elevated in the hippocampus. The lack of difference between the control group and the model group for this standard marker associated with the pathogenesis of AD with regard to AChE may be related to the fact that we measured the levels but not the enzyme activity, which is a limitation of the study and could be taken into account in further studies. We can also speculate that, in our model, the detected enzyme levels in the hippocampus were obtained before the formation of β-amyloid plaques, where the increased activity of AChE has been reported [34]. In addition, higher AChE activity in the vicinity of amyloid plaques has been shown to enhance Aβ aggregation, making it more toxic than Aβ fibrils [35]. Pretreatment with both the positive control melatonin and the compound **3c** reduced the model-induced increase in the Aβ and pTAU expression in the hippocampus of pin+Aβ_1-42_ rats to control levels. Interestingly, both the Aβ_1-40_ and Aβ_1-42_ impaired melatonin production and receptor signaling in cultured cells [36]. Therefore, melatonin deficiency in the pin+*icv*Aβ_1-42_ rat model may be associated not only with pineal ablation but also with decreased hormone production in other tissues.

The underlying molecular mechanism of melatonin and the compound **3c** was also reviewed and discussed. The accumulated evidence supports the hypothesis that melatonin may induce the non-amyloidogenic processing of APP while suppressing amyloidogenic processing, thus preventing the formation of neurotoxic Aβ oligomers and the further accumulation of Aβ in plaques [25]. The endogenous hormone stimulates the activity of alpha-secretases (ADAM10 and ADAM17) at the transcriptional level [6,37]. In contrast, melatonin suppresses amyloidogenic processing by downregulating beta-secretase (BACE1) (transcription, translation and enzyme activity). Furthermore, reduced BACE1 activity keeps the cholinergic system intact and suppresses neuronal damage and memory decline in parallel with the suppression of Aβ_40/42_ production [38,39].

The neuroprotective effects of melatonin may be mediated by both receptor-dependent and receptor-independent mechanisms [25]. The reduction in SIRT1 levels is negatively correlated with the duration of AD symptoms [40,41,42]. Melatonin promotes the expression of SIRT1 in primary neurons shortly after exposure for up to 24 h via a receptor-independent mode [43], suggesting that the upregulation of SIRT1 may promote the expression of ADAM10, which is involved in non-amyloidogenic processing, thereby leading to a reduced production of Aβ.

Melatonin can also exert its effects through MT_1_ and MT_2_ receptors, which are expressed jointly and individually in brain structures [23,44]. While the MT_1_ receptor is widely distributed in various tissues, the MT_2_ receptor has a more restricted distribution and is mainly found in the brain, including the hippocampus [23]. Melatonin, through its plasma receptors, induces ERK1/2 phosphorylation via various signaling pathways and activates transcription factors such as CREB [7]. Our results showed that, while melatonin up-regulated SIRT1 expression, the melatonin-related compound **3c** was ineffective, suggesting that the hybrid compound could mimic the hormone’s effects mainly through MT receptor activation. Furthermore, our results showed that, while melatonin and the compound **3c** reversed the model-related down-regulation of MT_1A_ receptors in the hippocampus, only the novel melatonin-like hybrid corrected the Aβ-induced reduced MT_1B_ receptors. The positive effect of the two pretreatments (melatonin and **3c**) on the pCREB expression suggests that the ERK1/2/CREB signaling is the underlying mechanism involved in the protective effects of the two drugs against Aβ formation via the activation of MT_1A_ receptors (melatonin and **3c**) and MT_2B_ (**3c**). However, the role of SIRT1 in the effects of melatonin, which may be be realized via a non-receptor mode, cannot be excluded.

## 4. Materials and Methods

The compound **3c** was synthesized according to procedure described previously in the literature [16] and the structure was proved by ^1^H, ^13^C NMR spectra and HRMS.

### 4.1. Animals

Adult male Wistar rats (250–300 g bw), purchased by the vivarium of the Institute of Neurobiology, BAS, were acclimatized in standard conditions: Plexiglas cages in groups of four; 12/12 cycled light/dark regime; average temperature at 22–23 °C; and food and water *ad libitum.* The experimental design was prepared in full accordance with the European Communities Council Directive 2010/63/E.U. and approved by the Bulgarian Food Safety Agency (research project: #347).

### 4.2. Experimental Design

Rats were randomly assigned to four groups (n = 7–10 rats/group) as follows: Group 1 (sham-veh-veh; a control sham-operated rats, i.p. injected after surgery with saline once a day for 14 days and *icv* infused with saline after a week), group 2 (pin-Aβ-veh; a model group with pinealectomy, i.p. injected after surgery with saline as controls and *icv* infused with Aβ_1-42_), group 3 (pin-Aβ-mel; a group with pinealectomy, i.p. injected after surgery with melatonin (10 mg/kg, i.p. for 14 days) and *icv* infused with Aβ_1-42_) and group 4 (pin-Aβ-**3c**; a group with pinealectomy, i.p. injected after surgery with 3c compound (10 mg/kg, i.p. for 14 days) and *icv* infused with Aβ_1-42_.

#### 4.2.1. Surgery and icv Injection of Aβ_1-42_

The surgical procedure was performed as described in our previous study [15], except the fact that pineal removal was performed a week prior to *icv* A*b*_1-42_ infusion. The sham-operated group was treated in the same way but without pinealectomy and with *icv* infusion of vehicle instead of Aβ_1-42_. The neurotoxic fibrils were prepared according to the procedure described in a previous study [15]. Briefly, the stock solution (1 µg/1 µL) was incubated for seven days at room temperature prior to infusion.

#### 4.2.2. Behavioral Test

##### Actimeter

Locomotion and diurnal rhythm changes were monitored for 26 h using an infrared actimeter (Bioseb, Paris, France). The first two hours were considered an adaptation period and were not included in the results. Each rat was tested in a sensor box (20 cm high, 45 cm wide and 45 cm long), measuring horizontal and vertical activity, distance and mean velocity. For the 24 h activity, cosinor analysis was performed to determine the mean amplitude, acrophase and mensor of the groups in relation to their rhythmic variations, together with the zero amplitude test.

##### Object Recognition Test (ORT)

The ORT was performed according to the procedures described in our previous study [45]. Briefly, after 24 h of habituation to an empty open field apparatus (50 × 50 × 50 cm), the rats were placed in it next to two identical plastic objects (referred to as “F”) for a 5 min (Training phase). After a sixty-minute interval, the rats were returned to the same box for the Testing phase. In this phase, one of the objects was replaced with a novel object (designated “N”) for 5 min. Throughout both the Training and Testing phases, careful observations were made, and the time spent and the number of sniffs of each object were recorded. Exploration time was quantified in seconds, and each object count (number of sniffs) was documented. The DI was then calculated using the formula: (N)/(N + F).

##### Object Location Test (OLT)

The OLT consists of several steps. During the adaptation phase, each animal was placed in a box (50 × 50 cm). The adaptation phase lasted 10 min, allowing the animals to become accustomed to their environment without being scored. After 24 h, the animals were transferred to a dimly lit test room 30 min before the start of the test. Two identical objects were placed individually in the same box, positioned in opposite corners. The animals were given 5 min to freely explore and interact with the objects. After 60 min, the animals were subjected to the test phase. One of the objects was moved to the opposite corner, creating both a familiar and a novel location. The animals’ behavior during this 5 min period was recorded. The duration (in seconds) of sniffing at the familiar and novel locations was recorded. The discrimination index was calculated using the formula: DI = (NO × 100)/(NO + FO). This index provided insight into the animals’ ability to distinguish between familiar and novel object locations. To maintain consistency and avoid any unwanted odors, the arms of the maze were wiped with vinegar or alcohol between each test.

##### Y-Maze Test

The Y-maze test consisted of a setup with three steel arms positioned at 120° angles to each other. Working memory was assessed in the first trial. Each rat was placed in the center of the apparatus and given 8 min to freely explore the three arms. Access to these arms was manually recorded by two persons blind to the schedule. SAB was calculated on the basis of visits to triads consisting of three arms. SAB was determined using a formula that took into account the number of arm entries and triads; alternation % = (number of correct entries × 100)/(total entries (N) − 2). In the second trial, conducted at least 5 days later, a pretest was performed by closing off one arm and allowing the rat to explore only two arms for 10 min. After a 30 min interval, the rat was placed in the arm opposite to the two arms initially explored. During this phase, the time spent in the two new arms and the number of entries into each arm were measured.

#### 4.2.3. Detection of Aβ, pTAU, AchE, pERK, pCREB and SIRT1 in the Homogenates from the Hippocampus

After the last memory test, the animals were euthanized by guillotine, and the left and right hippocampi were carefully isolated, snap-frozen in liquid nitrogen and stored in a refrigerator until further analysis (8 samples/group). Hippocampal samples were weighed and homogenized with HEPES buffer (20 mM HEPES; 1 mM EGTA; 210 mM Mannithol; 70 mM sucrose; pH 7.2) and protease inhibitor cocktail (100 mM PMSF, 100 mM NaF, 35 mM EDTA). The homogenates were centrifuged at 10,000× *g* for 5 min at 4 °C. The supernatants were used for the determination of total protein by the Bradford method followed by ELISA tests to measure Aβ, pTAU and AchE. The ELISA measurements were performed according to the manufacturer’s guidance for the specific kits (Elabscience Rat Aβ1-42, E-EL-R1402; Elabscience Rat pMAPT/pTAU (phosphorylated microtubule-associated protein tau), E-EL-R1090; Elabscience Rat AChE (Acetylcholinesterase), E-EL-R0355).

##### Statistical Analysis

The 24 circadian rhythms of motor activity in the actimeter were assessed by two-way repeated ANOVA. Two-way ANOVA was applied to determine the diurnal variations in horizontal and vertical activity, distance and mean velocity. Memory and biochemical data were evaluated by one-way ANOVA. The Shapiro–Wilk post hoc test was used for homogenously distributed data, and the Kruskal–Wallis test on ranks followed by the Mann–Whitney U test were applied for non-parametric data. The significant level was set at *p* < 0.05.

## 5. Conclusions

Melatonin supplementation facilitated the non-amyloidogenic signaling via non-receptor (histone deacetylase sirtuin 1, SIRT1) and receptor MT_1A_-related signaling (ERK1/2/CREB) in a rat model of pin+icvAβ_1-42_. The novel hybrid analogue **3c** positively affected the pin+icvAβ_1-42_-induced hippocampus-dependent spatial memory impairment by stimulating MT_1A_/MT_2B_ receptors, which trigger ERK1/2/CREB non-amyloidogenic signaling in rats. Melatonin supplementation or other interventions targeting the melatoninergic system may hold promise as potential strategies for treating aspects of AD. Given that pineal gland dysfunction may be only one of many causes of AD pathogenesis, further research is needed to fully understand the underlying mechanisms and develop practical therapeutic approaches.

## Data Availability

The data that support the findings of this study are available from the corresponding author upon reasonable request.

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
