# Peer review of "Protective Effect of the Novel Melatonin Analogue Containing Donepezil Fragment on Memory Impairment via MT/ERK/CREB Signaling in the Hippocampus in a Rat Model of Pinealectomy and Subsequent Aβ1-42 Infusion"

_ijms, 2024, doi:10.3390/ijms25031867_

Round 1

Reviewer 1 Report

Comments and Suggestions for Authors

In the manuscript entitled “Protective Effect of the Novel Melatonin Analogue, Containing 2 Donepezil Fragment on Memory Impairment via 3 MT/ERK/CREB Signaling in the Hippocampus in a Rat Model 4 of Pinealectomy and Subsequent Aβ1-42 Infusion’’ by Jana Dimitrova Tchekalarova and her colleagues, investigated the neuroprotective properties of melatonin analogue in a rat model of pinealectomy and subsequent peptide beta-amyloid inject.

 The manuscript is interesting, however important modifications would need to be made to be considered valid and it cannot be accepted in this form.

The manuscript is written in a confusing way and quite difficult to read. The text must be rewritten, and major editing work should be done to unify the text and make it clear not only to physiologists.

 The introduction should be clearer and more homogeneous, and some concepts should be extended, for example, the data about the reduction of pineal gland in Alzheimer disease should be reported. Parts of the text currently under discussion could be moved to the introduction. Very clear scheme of whole study design with timeline from Fig 10 can be moved to the end of introduction.

 The results are quite confusing, they should be rewritten and explained better and with greater accuracy to be truly convincing:

-        Figure 2 legend contains obviously erroneous reference to Fig. 1.

-        Cosinor data should be explained

 A flowchart with the main effects obtained and key pathways affected by melatonin replenishment should be added to the discussion.

Comments on the Quality of English Language

The text is significantly overloaded with details, making it difficult to comprehend

Author Response

Thank you for the careful evaluation of our manuscript. We have revised the manuscript taking into account the suggested modifications. All changes in the MS are highlighted by track changes.

Reviewer #1

Point #1 The manuscript is written in a confusing way and quite difficult to read. The text must be rewritten, and major editing work should be done to unify the text and make it clear not only to physiologists. The introduction should be clearer and more homogeneous, and some concepts should be extended, for example, the data about the reduction of pineal gland in Alzheimer disease should be reported. Parts of the text currently under discussion could be moved to the introduction.

Response: We agree with this remark. The text has been duly rectified in line with this feedback. In response to the Reviewer's suggestion, we have revised the Introduction to enhance clarity and logical coherence. Information pertaining to the diminished function of the pineal gland in Alzheimer's disease has been incorporated into this section. Additionally, certain paragraphs unrelated to the discussion of results have been relocated from the Discussion section to the Introduction, spanning from line 265 (pre-revision) to line 70 (post-revision).

Point #2:  Very clear scheme of whole study design with timeline from Fig 10 can be moved to the end of introduction.

Response: We concur with the reviewer's suggestion and have relocated the aforementioned figure with the timeline to the end of the Introduction, specifically in the last paragraph.

Point #3:  The results are quite confusing, they should be rewritten and explained better and with greater accuracy to be truly convincing:

-        Figure 2 legend contains obviously erroneous reference to Fig. 1.

-        Cosinor data should be explained

Response: We appreciate the reviewer's insightful comment. The description of the results has been thoroughly reviewed and appropriately adjusted for improved clarity. References to Figure 3 (formerly Fig. 2) in the text have been corrected. Regarding Cosinor data, a table containing parameters related to circadian rhythms for each group has been inserted, and the corresponding results have been elaborated upon in the text. Also, the cosinor anaylis was described in Method section.

Point #4:   A flowchart with the main effects obtained and key pathways affected by melatonin replenishment should be added to the discussion.

Response: We appreciate this suggestion. Indeed, the inclusion of a flowchart depicting the main effects and pathways will enhance the clarity and ease of understanding of the main findings. A flowchart has been created and inserted into the Discussion section (Scheme 1).

Reviewer 2 Report

Comments and Suggestions for Authors

Submitted manuscript presents sound data revealing that synthesized melatonin derivative displays beneficial effects  on memory decline and hippocampal rise of Aβ1-42 and pTAU. The design of the study and conclusions are appropriate. However, certain changes are needed prior to publication.

Comments:

1) Please, use consistent statistical approach, i.e. consider the data with p<0.05 significant and either present everywhere the precise p value or use standard three steps <0.05, <0.01 and <0.001.

2) In reference to above, please correct the unclear statement (lines 131-132) "the sub-chronic treatment with melatonin or 3c showed a tendency to reverse the impaired memory recognition (p > 0.05 vs. sham-veh-veh group) and (Fig. 4B)"

if there is no statistical significance there is no effect, therefore, please refrain from discussing trends

2) please, indicate precisely the number of animals in studied groups instead if stating N= 8-10

Comments on the Quality of English Language

Please, check the manuscript for misspelled words

Author Response

Thank you for the careful evaluation of our manuscript. We have revised the manuscript taking into account the suggested modifications. All changes in the MS are highlighted by track changes.

Reviewer #2

Point #1 Please, use consistent statistical approach, i.e. consider the data with p<0.05 significant and either present everywhere the precise p value or use standard three steps <0.05, <0.01 and <0.0 01.

Response: We concur with this pertinent observation from the reviewer. In the revised version of the manuscript, we have uniformly presented data from the statistical analysis, applying the standard three steps of p-values.

Point #2: In reference to above, please correct the unclear statement (lines 131-132) "the sub-chronic treatment with melatonin or 3c showed a tendency to reverse the impaired memory recognition (p > 0.05 vs. sham-veh-veh group) and (Fig. 4B)" if there is no statistical significance there is no effect, therefore, please refrain from discussing trends

Response: The text mentioned above by the Reviewer was corrected to outline the lack of effect of the tested drugs.

Point #3 Please, indicate precisely the number of animals in studied groups instead if stating N= 8-10

Response: The number of animals was precisely described in the new version of the text.

Point #4: Comments on the Quality of English Language Please, check the manuscript for misspelled words

Responses: The final version was carefully checked again for grammar. We are thankful for this note.

Reviewer 3 Report

Comments and Suggestions for Authors

The manuscript by Jana Dimitrova Tchekalarova et al demonstrated that therapeutic effect of the new melatonin analog on Memory Impairment in the Hippocampus in the rat of pinealectomy and subsequent Aβ1-42 Infusion. The study appears to be of interest, the general purpose of this study is clear. However, I think the experiments have some concerns. In my opinion, therefore, this manuscript is recommended for publication in the paper after throughout a few modifications.

Comment #1: More data

Alzheimer’s disease (AD) is a progressive neurodegenerative disease most often associated with memory deficits and cognitive decline. Neuronal cell death is one of the most typical pathological hallmarks of neurodegenerative diseases. Therefore, it is the most important aspect of Alzheimer's and dementia medications whether it has a protective effects on neuronal death in the hippocampus. For example, in the immunostaining assay, did the TUNEL-positive apoptotic cells decrease in the analog treated model rats? The authors should modified this point in method section of the revised manuscript

Comment #2: How did you agglutinate Aβ1-42? The authors should modified this point in method section of the revised manuscript

Author Response

Thank you for the careful evaluation of our manuscript. We have revised the manuscript taking into account the suggested modifications. All changes in the MS are highlighted by track changes.

Reviewer #3

Point #1 The manuscript by Jana Dimitrova Tchekalarova et al demonstrated that therapeutic effect of the new melatonin analog on Memory Impairment in the Hippocampus in the rat of pinealectomy and subsequent Aβ1-42 Infusion. The study appears to be of interest, the general purpose of this study is clear. However, I think the experiments have some concerns. In my opinion, therefore, this manuscript is recommended for publication in the paper after throughout a few modifications.

More data

Alzheimer’s disease (AD) is a progressive neurodegenerative disease most often associated with memory deficits and cognitive decline. Neuronal cell death is one of the most typical pathological hallmarks of neurodegenerative diseases. Therefore, it is the most important aspect of Alzheimer's and dementia medications whether it has a protective effects on neuronal death in the hippocampus. For example, in the immunostaining assay, did the TUNEL-positive apoptotic cells decrease in the analog treated model rats? The authors should modified this point in method section of the revised manuscript

 Response: We fully agree with this opinion of the Reviewer! Indeed, further we plan to continue our research on immunostaining assays to evaluate the plausible neuroprotective effect of the hybrid compound 3c in this rat model of sporadic AD. The neuronal loss in this model was recently verified and reported in our previous study (Ilieva K, Atanasova M, Atanasova D, Kortenska L, Tchekalarova J. Chronic agomelatine treatment alleviates icvAβ-induced anxiety and depressive-like behavior through affecting Aβ metabolism in the hippocampus in a rat model of Alzheimer's disease. Physiol Behav. 2021 Jul 6:113525. doi: 10.1016/j.physbeh.2021.113525).

 Point #2 Comment #2: How did you agglutinate Aβ1-42? The authors should modified this point in method section of the revised manuscript

Response: We have described the approach for preparing the Aβ1-42 aglutinations in our previous studies (Ref. 15 in the present study). Following the Reviewer’s comment details on preparation of the neurotoxic fibrils have been reinserted in the text again (line  439).

Reviewer 4 Report

Comments and Suggestions for Authors

Q1 line 48 the orexin receptor,

It sounds to me like an abrupt insertion here about the orexin receptor. Would it be good if you can introduce this receptor with some words, especially the relationship between this receptor and melatonin to help your readers to know more about your story?

Q2: Figure 7A

It seems to me that only two point were extremely higher in the column of the pin-Aβ-veh. Most points were distributing below 60, and the data seem unlikely to show statistical differences when comparing with the sham-veh-veh.

Q3 line 54-56: “The hormone’s primary function is the regulatory control of the circadian rhythms in the body, including synchronization of the sleep-wake cycle and promotion of sleep. It is hypothesized that melatonin deficiency might be a critical trigger predisposing to AD pathogenesis”

Pineal gland dysfunction has been found to link to AD, but, obviously, not all patients with AD have pineal gland dysfunction. If patients with AD showing a good response to melatonin, but their pineal glands are intact and functions are good, and are irrelevant to melatonin deficiency, how do you explain the effects of melatonin or your 3c in such patients with AD?

Literature shows that melatonin may have anti-inflammatory effects (doi: 10.1111/j.1749-6632.2000.tb05402.x.). Therefore, the effects of melatonin may be far more we can understand. I am glad that you have addressed this point in your discussion.

Another point deserves the authors to discuss is the gut may produce several hundred-fold more melatonin than the pineal gland generatesdoi: 10.1007/BF01923948.). Is it a problem to affect your results?

Q4: line 54-55 The hormone’s primary function is the regulatory control of the circadian rhythms in the body, including synchronization of the sleep-wake cycle and promotion of sleep

In those statements, you suggest that one main function of melatonin is the regulatory control of the circadian rhythms. Therefore, you cut off the pineal gland as a resort of creating an animal with melatonin deficiency. However, we know that the pineal gland is present in almost all vertebrates, but is absent in protochordates. If your statements are right, such organisms may have circadian problems?

I found a paper which did not agree with this viewpoint (https://doi.org/10.1016/0022-0981(90)90168-C).

In combination with Q3, I would suggest the authors to address that “the defect of pineal gland” is only one of many causes for the pathogenesis of AD, and AD related circadian rhythms dysfunction. 

Q5: Need an illustration of signal transduction pathways to explain the findings of Fig 9.

Author Response

Thank you for the careful evaluation of our manuscript. We have revised the manuscript taking into account the suggested modifications. All changes in the MS are highlighted by track changes.

Reviewer #4

Point #1 Q1 line 48 the orexin receptor, It sounds to me like an abrupt insertion here about the orexin receptor. Would it be good if you can introduce this receptor with some words, especially the relationship between this receptor and melatonin to help your readers to know more about your story?

Response: The selective antagonist of orexin, Suvorexant, differs significantly from melatonin, except for its ability to alleviate disturbed sleep-wake circadian patterns in the early stages of Alzheimer's disease (AD). We administered this clinical sample to address impaired sleep-wake cycles at the outset or before the onset of AD symptoms, employing an approach aimed at mitigating or suppressing the development of AD-related pathogenesis. To simulate the condition of sleep-wake circadian pattern impairment, our model involved pinealectomy and concurrent treatment with melatonin to regulate the disturbed balance and subsequent Abeta-related pathology.

Point #2 Q2: Figure 7A It seems to me that only two point were extremely higher in the column of the pin-Aβ-veh. Most points were distributing below 60, and the data seem unlikely to show statistical differences when comparing with the sham-veh-veh.

Response: Indeed, among the n = 8 in each group, only two data points were higher and one lower than the average for the Abeta level: (sham-veh-veh – Abeta: 44.63±1.76; pin-Abeta-veh – 60.26±7.03; pin-Abeta-mel – 47.77±4.01; pin-Abeta-3c – 53.29±4.55). One-way ANOVA (parametric) test was used suggesting a main group effect with a post hoc result demonstrating a significant difference between the pin-Aβ1-42-veh and sham-veh-veh group (p < 0.05).

Point #3 Q3 line 54-56: “The hormone’s primary function is the regulatory control of the circadian rhythms in the body, including synchronization of the sleep-wake cycle and promotion of sleep. It is hypothesized that melatonin deficiency might be a critical trigger predisposing to AD pathogenesis” Pineal gland dysfunction has been found to link to AD, but, obviously, not all patients with AD have pineal gland dysfunction. If patients with AD showing a good response to melatonin, but their pineal glands are intact and functions are good, and are irrelevant to melatonin deficiency, how do you explain the effects of melatonin or your 3c in such patients with AD?

Response: The melatonin deficiency is considered as one of the hypothesis for predisposition to AD pathogenesis. The development of sporadic AD increase with aging when the function of the pineal gland is significantly reduced. Also, clinical reports showed that AD patients had low level of melatonin in CSF, serum and urine suggesting an impaired function of the pineal gland. Also, reports demonstrated a positive relationship among the low melatonin level in AD brain, cognitive dysfunction in AD patients and the reduced pineal gland volume (reviewed in Song et al., 2019). Interestingly, the expression of melatonin receptor such as MT2 was decreased in the hippocampus of AD patients (Savaskan E et al., 2002), suggesting that the novel melatonin analogue 3c might be a good candidate and more potent than melatonin against consequences related to melatonin deficiency and AD pathogenesis.

Point #4 Literature shows that melatonin may have anti-inflammatory effects (doi: 10.1111/j.1749-6632.2000.tb05402.x.). Therefore, the effects of melatonin may be far more we can understand. I am glad that you have addressed this point in your discussion.

Response: We fully agree with this opinion of the Reviewer! Indeed, further we are planning to continue our research on anti-inflammatory potential activity of the 3c compound and melatonin applied as a referent drug in the same model of melatonin deficiency with a concomitant icv Abeta infusion. The activated glia (microglia and astrocytes) and pro-inflammatory cytokines will be evaluated.

Point #5 Another point deserves the authors to discuss is the gut may produce several hundred-fold more melatonin than the pineal gland generates ( doi: 10.1007/BF01923948.). Is it a problem to affect your results?

Response: The tissues with the highest metabolic activity, like the brain, skin, and gut, are the most critical parts of the body where the indoleamine is also synthesized (DOI 10.1007/978-1-61779-313-4_7; Tan D.X et al., 2007). Further, results from animals and humans reveal that extrapineal melatonin levels can exceed the blood levels of the hormone (doi: 10.2174/157015910792246182; Tan D.X et al., 1999). Both glandular (pineal) and cellular sources of detected melatonin in the brain tissue contribute to the higher level in cerebrospinal fluid than other places (Brzezinski A et al., 1987, Tricoire H et al., 2002). However, although many other extrapineal tissues are sources of melatonin, its hormonal activity primarily involves fine-tuning endocrine and other internal “clock” signals synchronized with the external “clock” of the light-dark cycle. Thus, the role of melatonin to synchronize circadian rhythms and wake-sleep cycle is closely related to its secretion from the pineal gland which dysfunction as we mentioned before might underline one of the mechanism underlying predisposition to development of AD.

Point #6 Q4: line 54-55 The hormone’s primary function is the regulatory control of the circadian rhythms in the body, including synchronization of the sleep-wake cycle and promotion of sleep. In those statements, you suggest that one main function of melatonin is the regulatory control of the circadian rhythms. Therefore, you cut off the pineal gland as a resort of creating an animal with melatonin deficiency. However, we know that the pineal gland is present in almost all vertebrates, but is absent in protochordates. If your statements are right, such organisms may have circadian problems? I found a paper which did not agree with this viewpoint (https://doi.org/10.1016/0022-0981(90)90168-C).

 Response: Well, that is an interesting suggestion but I am afraid this issue is not related to the topic of our model of AD and the potential treatment approaches related to non-amyloidogenic mechanism to suppress AD pathogenesis.

Point #7 In combination with Q3, I would suggest the authors to address that “the defect of pineal gland” is only one of many causes for the pathogenesis of AD, and AD related circadian rhythms dysfunction. 

 Response: We agree with this suggestion of the Reviewer and pointed out this state in the final sentence in Conclusion.

Point # 8 Need an illustration of signal transduction pathways to explain the findings of Fig 9.

 Response: We inserted a scheme of suggested mechanism of action of melatonin and 3c compound in the model of pin + icvAβ1-42 (Scheme 1) in the end of the first paragraph of Discussion.

Round 2

Reviewer 3 Report

Comments and Suggestions for Authors

I have no concerns about publish.